# Precise Positioning in Nitrogen Fertility Sensing in Maize (*Zea mays* L.)

**DOI:** 10.3390/s24165322

**Published:** 2024-08-17

**Authors:** Tri Setiyono

**Affiliations:** Louisiana State University Agriculture Center, School of Plant, Environmental and Soil Sciences, Baton Rouge, LA 70803, USA; tsetiyono@agcenter.lsu.edu; Tel.: +1-225-578-3135

**Keywords:** GNSS, RTK, maize, nitrogen, UAV, remote sensing, multispectral, abiotic stress, excessive water, field crops

## Abstract

This study documented the contribution of precise positioning involving a global navigation satellite system (GNSS) and a real-time kinematic (RTK) system in unmanned aerial vehicle (UAV) photogrammetry, particularly for establishing the coordinate data of ground control points (GCPs). Without augmentation, GNSS positioning solutions are inaccurate and pose a high degree of uncertainty if such data are used in UAV data processing for mapping. The evaluation included a comparative assessment of sample coordinates involving RTK and an ordinary GPS device and the application of precise GCP data for UAV photogrammetry in field crop research, monitoring nitrogen deficiency stress in maize. This study confirmed the superior performance of the RTK system in providing positional data, with 4 cm bias as compared to 311 cm with the non-augmented GNSS technique, making it suitable for use in agronomic research involving row crops. Precise GCP data in this study allow the UAV-based Normalized Difference Red-Edge Index (NDRE) data to effectively characterize maize crop responses to N nutrition during the growing season, with detailed analyses revealing the causal relationship in that a compromised optimum canopy chlorophyll content under limiting nitrogen environment was the reason for reduced canopy cover under an N-deficiency environment. Without RTK-based GCPs, different and, to some degree, misleading results were evident, and therefore, this study warrants the requirement of precise GCP data for scientific research investigations attempting to use UAV photogrammetry for agronomic field crop study.

## 1. Introduction

The term global navigation satellite system (GNSS) refers to the collection of satellite navigation systems and their augmentations to provide three-dimensional position information to users with the appropriate receiving equipment [1]. Augmentation of GNSS data is critical to mitigate position inaccuracy due to satellite positioning, clock errors, and atmospheric noise [2]. Currently popular GNSS augmentation technology in land survey includes real-time kinematics (RTK) involving signal correction by reference stations [3]. Accurate and robust performance of RTK [4] and increased availability of this differential correction technology allows widespread applications of GNSS in agriculture [5], particularly for precision agricultural operations such as tillage, planting, and weed control [6,7,8].

Unmanned aerial vehicle (UAV) photogrammetry is gaining popularity in agricultural research due to its high observation resolutions compared to satellite remote sensing [9]. When a highly accurate topographic map is required, particularly for time-lapse analysis, the positional information of the UAV remote sensing data must be corrected and optimized [9]. The use of ground control points (GCPs) is one of the most used methods to correct the positional information of UAV imageries [10]. The geographic coordinates of GCPs are recorded using GNSS technology and ideally using precise augmentation such as RTK technology.

Recognizing the trend of continued widespread interest in UAV photogrammetry applications in agricultural research, this study aims to document the merit of GCP coordinates determination using precise positioning involving GNSS RTK in the context of N fertility study in maize (*Zea mays* L.). Absolute positioning relying on a single receiver such as the case of individual camera data acquired by UAV is corrupted by unmitigated errors inherent in satellite positioning, such as satellite orbit errors, clock errors of both satellites and receivers, atmospheric (ionospheric and tropospheric) errors, multipath errors, and receiver noise [2]. It is therefore important to assess how the precision positioning technology affects UAV remote sensing results in the context of monitoring nitrogen deficiency stress impacting non-N-fixing field crops such as maize. This assessment of precise GCP data contribution to UAV photogrammetry is useful to justify whether the additional workload of GCP management is worthwhile in the entire UAV-based aerial mapping processing workflow and to ensure successful applications of sensing technology in agronomic research. Earlier efforts, for example, were taken to reduce the GCP requirement to only four corners of a target location without sacrificing the quality of the resulting maps [11]. While GNSS augmentation with RTK has been widely adopted in agricultural operations [7], there are still uncertainties in the literature in terms of its merit in research applications, particularly for UAV-based mapping. The objective of this study is to provide a scientific report of the performance of GNSS augmentation with RTK in the context of mapping for agronomic research.

## 2. Materials and Methods

### 2.1. Comparative Analysis of Position Precision of RTK versus Non-Augmented GPS

The GNSS RTK used in this study was EMLID RS2 (EMLID Tech Kft., Budapest, Hungary) (Figure 1). An assessment was conducted at the LSU AgCenter Central Research Center, Baton Rouge, LA, USA, on 11 November 2023 to evaluate the positional precision of the RTK system used in this study by comparing it against a non-augmented ordinary GPS device (Garmin GPSMap65, Garmin, Olathe, KS, USA). The RTK system consisted of rover and base units (Figure 1). The base and rover RTK units were set and controlled with the ReachView application (EMLID Tech Kft., Budapest, Hungary) installed on a Nautiz X6 Android device (Handheld Group AB, Lidköping, Sweden).

The absolute position of RTK base coordinates was determined using the US National Oceanic Atmospheric Administration (NOAA) Online Positioning User Service (OPUS) (geodesy.noaa.gov (accessed on 10 May 2022)) (Figure 2). This publicly available web-based service utilizes US National Geodetic Survey (NGS) computers and the software package called Program for the Adjustment of GPS Ephemerides (PAGES) version 5.00 to compute highly accurate coordinates from the provided GNSS data for the base location in this study. The static processing option of OPUS (OPUS-S) was used, requiring GNSS data collection for more than 2 h, and involved data from the three nearest Continuously Operating Reference Stations (CORSs) and a double-differenced, ionospheric-free mathematical model [12].

Raw data logging for submission to NOAA OPUS was configured at 6 s, averaging with the raw data format of RINEX (the IGS Receiver Independent Exchange) 3.03, with a logging interval of 30 s for 3 h of continuous data collection. The resulting raw data along with antenna height information were submitted to the NOAA OPUS system (https://www.ngs.noaa.gov/OPUS/ (accessed on 10 May 2022)). The corrected output coordinates of the base station received from NOAA OPUS via e-mail were in the Coordinate Reference System (CRS) EPSG: 3745 (NAD83 HRN/UTM Zone 15N) (Figure 2b). These coordinates were transformed into the EPSG: 4326 (Geographic) CRS using the Geospatial Data Abstraction Library (GDAL) [13] and QGIS [14], such that the data could be entered manually to the EMLID RS2 RTK base unit when it was set up to provide kinematic corrections to the rover unit for determining target coordinate data. During data collection, the RTK-based unit was set precisely at the marked coordinate point (0.5 cm diameter plastic marker). This precise RTK base unit setup was facilitated by a tribrach equipped with an optical plummet for viewing the target marker on the ground and making fine adjustments to the base accordingly. This tribrach plate attachment served as an adapter between the RTK-based unit and the survey tripod (Figure 1). The RTK rover unit was mounted on survey rod with a Nautiz X6 Android device attached. Communication between the RTK base and rover units was facilitated with a long-range (LoRA) radio frequency at 902.5 MHz. The application of a GPSMap65 device for comparative analysis against RTK involved waypoint averaging to at least 90% confidence intervals. The coordinates data taken with RTK and GPSMap65 were used in UAV data processing (Figure 3, Section 2.2).

The position accuracy assessment involved position measurements of 13 points along the external perimeter of blocks 31, 32, and 33 at the LSU AgCenter Central Research Center (Figure 4). The collected coordinate data from RTK and GPS units were first transformed into EPSG: 3745 (NAD83 HRN/UTM Zone 15N) using the Geospatial Data Abstraction Library (GDAL) [13] and QGIS [14] and then subjected to distance computation against the benchmark coordinate data that were determined earlier using RTK EMLID RS2 on 3 May 2023. The distance between the test coordinates and the benchmark position (d, m) was computed based on the Pythagorean theorem solution for hypotenuses as follows:(1)$$d=\sqrt{{\left(X-X_{0}\right)}^{2}+{\left(Y-Y_{0}\right)}^{2}}$$
where X is the easting value of the test position (m), X_0_ is the easting value of the benchmark position (m), Y is the northing value of the test position (m), and Y_0_ is the northing value of the benchmark position (m). The closer the d value to 0, the more accurate the test coordinate measurement. The UAV DJI Matrice 300 aircraft (DJI, Shenzhen, China) equipped with D-RTK2 (DJI, Shenzhen, China) was flown on 6 Jun 2023 with a 12 MP DJI H2O sensor (DJI, Shenzhen, China) at the altitude of 65 m above ground to capture a background RGB map for the comparative assessment of the positional accuracy of RTK vs. GPS. This map was also used to generate benchmark coordinate data used for Equation (1).

### 2.2. RTK-GCP Merit Evaluation in UAV Photogrammetry

The evaluation of RTK’s application in UAV photogrammetry is based on maize field crop trials at block H1 of the Red River Research Station in Bossier City, LA, USA, in 2022 (Figure 5). Maize crops, cultivar DKC-70-27, were planted on 28 March 2022 at a seeding rate of 78,085 seeds ha^−1^ with row planters on beds with furrows to facilitate water drainage in the field, with a row spacing of 1.02 m. The soil type was Latanier Clay with 0–1% slope. The four-replication split-plot experiment consists of the main plot of water environments (W1: non-excessive vs. W2: excessive) and fertilizer N rates as sub-plots. The four N rates, 0, 45, 180, and 224 kg ha^−1^, were applied on 27 April 2022. The excessive amount of water of 127 mm was applied using poly pipes for 10 h on 29 April 2022. Such excessive water application was intended to simulate extreme rain soon after N fertilizer application. Yield data for the maize crops were collected on 18 August 2022 using a combine harvester equipped with a GrainGage harvest master (Juniper Systems, Logan, UT, USA).

Five UAV flights were carried out 32 (29 April), 51 (18 May), 67 (3 June), 87 (23 June), and 106 (12 July) days after planting (DAP) using a DJI Matrice 300 UAV aircraft (DJI, Shenzhen, China) with a multispectral MicaSense Red-Edge-MX sensor (MicaSense Inc., Seattle, WA, USA) and supported with D-RTK2 (DJI, Shenzhen, China), with a flight altitude 40 m above ground. The type of spectral band and center wavelength for the MicaSense Red-Edge-MX sensor used in this study were as follows: blue (475 nm), green (560 nm), red (668 nm), red edge (717 nm), and NIR (840 nm). Orthomosaic images were generated using Pix4Dmapper (Pi4D S.A., Prilly, Switzerland), the Geospatial Data Abstraction Library (GDAL) [13], and QGIS [14]. 

The map generation with Pix4DMapper was conducted either involving RTK-based GCPs (+R) or without them (−R). For the treatment of non-RTK-based GCPs, the coordinates of GCPs were determined using a non-augmented GPS device. During UAV photogrammetry processing in Pix4DMapper, GCP coordinates data were supplied in the process at the initialization step (Figure 3) involving the raw UAV data containing GCP images, followed by re-optimization of camera positions and the internal camera parameters. The remote sensing index, the Normalized Difference Red-Edge Index (NDRE) [15], was used as main indicator of crop response to the N fertilizer applications and excessive water treatment:NDRE = (rNIR − rRE)/(rNIR + rRE) (2)
where rNIR is the reflectance of the near-infrared band (840 nm) and rRE is the reflectance of the red-edge band (717 nm).

The NDRE values for the RTK merit assessment were analyzed using two different approaches. The first approach was based on the entire treatment plot boundaries to obtain maize crop response to the treatments as captured using UAV photogrammetry. In this first approach, UAV photogrammetry is expected to capture the overall expression of maize crop canopy cover in response to the changes in N fertility dynamics [16]. In the second approach, NDRE value extraction was contained within a narrow section of 0.3 m along the center of the planted maize rows. In this second approach, UAV remote sensing sensor data are expected to capture the physiological basis of the dynamics of canopy cover observed in the first approach, as changes in N fertility under the environmental modifications in this study were expected to affect the canopy photosynthesis capacity, given the sensitivity of NDRE to canopy chlorophyll conditions [17]. Statistics data of NDRE by treatment using the above two approaches were extracted using zonal statistics in QGIS [14].

## 3. Results

The superior precision of RTK compared to ordinary GPS was evident from visual data of the 12 marker points (all test points except for the one marked by the arrow in Figure 4) along the reference boundary (combined boundaries of blocks 31, 32, and 33). As seen in Figure 4, these test points taken by the RTK system were relatively closer to these reference boundaries as compared to the points taken using GPS, and their spatial distribution along the same axis resembles straight lines. The distance computation involving Equation (1) established numerical evidence of the substantial differences in the precision of the point data taken with the RTK system (precision of 4.05 cm ± 2.835 cm) versus GPS (precision of 3.11 m ± 0.498 m). The 13th marker point was purposely not placed along the boundary lines of these research blocks and was thus excluded from the visual assessment, but it was included in the numerical assessment of the positional accuracy of spatial point determination using RTK vs. non-augmented GPS. Detailed results with bias data for x and y are shown in Table 1. Coordinate bias results from other studies are also presented in Table 1. 

The maize crop response to N fertilizer treatments and, to some degree, the excessive water treatment soon after fertilizer application was visually evident by the aerial map of NDRE at 67 DAP (Figure 5). Maize crops receiving a higher amount of N fertilizer showed higher NDRE values as compared to those receiving lower N rates. Low NDRE values were also observed in crops receiving excessive water treatment. This aerial map was generated through the UAV photogrammetry process involving the use of RTK-based GCP data. In the next assessment, time-series data of NDRE by treatments will be compared in detail, particularly comparing the results obtained when GCP data were established with RTK vs. non augmented GPS.

The first RTK-GCP merit assessment was based on approach 1 (refer to Section 2.2) involving zonal statistics computed using the entire plot boundaries for each treatment. The presentation of these results, which are identical for the next assessment using approach 2, are as follows. The date of measurements and NDRE values represent the X and Y axes in each individual graph panel (Figure 6 and Figure 7), respectively. The blue solid and red dashed lines in the graph represent the W1 (control) and W2 (excessive) water treatments, respectively. Vertically, the graph panels are arranged from highest (topmost) to lowest (bottom most) N treatments. Horizontally, the graph panels to the left represent UAV photogrammetry results involving the use of RTK-GCP (+R), whereas the graph panels to the right represent UAV photogrammetry results relying on non-augmented GPS for GCP coordinates establishment (−R). 

In general, the time-series graph displays increasing NDRE values with the progression of DAP until the peak observed at 87 DAP, as the last NDRE values indicate a decline in this remote sensing index as maize crops mature (Figure 6). These graphs also confirmed the visual clues shown in Figure 5 regarding the trend of higher NDRE in the case of plots receiving higher fertilizer N rates. The left graph panels in Figure 6a–d demonstrate the detailed trend of lower NDRE exhibited by plots receiving excessive water treatment soon after fertilizer applications, with the differences becoming more obvious at lower N rates. Plots receiving lower N rates were characterized by lower NDRE slopes toward the peak points (Figure 6a–d). As shown in the left panels in Figure 6, the reduction in the NDRE slopes toward the peak points were more evident under the excessive water treatments. 

The following are observations of the results obtained when non-augmented GPS was used for GCP coordinates establishment (Figure 6e–h). In the case of the highest N-rate treatment (Figure 6e), there is a similar trend of overall lower NDRE for W2 compared to W1, except for the overall lower NDRE values compared to the case in Figure 6a. In the case of the 180 kg N ha^−1^ treatment, at 57 and 67 DAP, an exaggeration of the impact of excessive water treatment was observed, followed by a misleading suggestion of greater NDRE values under the W2 treatment at 87 DAP as the peak of NDRE for W1 plots shifted to 67 DAP (Figure 6f). Such phenomena were also the case with the 45 kg N ha^−1^ treatment, with an exaggeration of the impact of excessive water treatment at 57 and 67 DAP, while the data suggested higher NDRE values under W2 compared to W1 treatments at 87 DAP. With the 0 kg N ha^−1^ treatment, the results were characterized with a higher-than-usual standard error of NDRE at 87 DAP and an unusual inflection of NDRE between 57 and 87 DAP for the W1 treatment.

In the second RTK-GCP merit assessment, the approach involved constraining zonal statistics for NDRE data extraction to a narrow 0.3 m along the center of the planted crop rows (refer to Section 2.2, approach 2). The results are shown in Figure 7. Results involving the use of RTK-based GCPs (Figure 7a–d) indicate higher NDRE values as compared to the NDRE values in Figure 6a–d. The results suggest a similar trend of reduced slope of NDRE progression over time for maize crops receiving lower N fertilizer rates, with the more evident of such trends being exhibited by the maize crops receiving excessive water treatment. Similar misleading cases were evident in the case of the second approach for UAV photogrammetry results relying on non-augmented GPS for GCP data (Figure 7e–h), with lower NDRE values for maize crops receiving 224 kg N ha^−1^ compared to results using RTK-GCPs (Figure 7a vs. Figure 7e), an exaggerated excessive water response for maize crops receiving 180 and 45 kg N ha^−1^ (Figure 7f,g), and an unusual NDRE inflection between 57 and 87 DAP as well as a higher-than-usual standard error of NDRE at 87 DAP for maize crops receiving 0 kg N ha^−1^ (Figure 7h).

The relationship between NDRE at 87 DAP and maize yield in this study is shown in Figure 8. The results were based on UAV photogrammetry involving RTK-GCPs. The linear regression equation described maize yield as a function of NDRE, with an R^2^ of 0.9097.

## 4. Discussion

The relatively low-cost RTK system used in this study, EMLID RS2 (retail price of less than USD 6000 for the base and rover units, as of December 2023), exhibited satisfactory cm level precision for coordinate determinations and superior performance as compared to a non-augmented GPS device (such as Garmin GPSMap 65) (Figure 4). This assessment provides evidence of the significant contribution of GNSS error mitigation with a carrier-based positioning approach in RTK compared to the simpler satellite-based pseudorange augmentation approach in ordinary GPS devices [3]. Even the poorest RTK result in this study with an error of 2.22 cm was still considerably more precise than the best results achieved with ordinary GPS at 1.78 m. The positional precision achieved with the RTK system in this study (an average of 4.05 cm) is comparable to results reported by others ([18,19,20], see Table 1). In the context of typical agronomic row crops, with research experiments involving crop row spacing of approximately 0.96 m, the ordinary GPS tested in this study (with an average precision of 3 m) is not ideal, as its precision was practically 300% more than the required value. On the other hand, the coordinate data determined by the RTK system used in this study were approximately 1.3% within the overall range of target row spacing.

The UAV-based NDRE map in this study can effectively characterize maize crop responses to N fertilizer amendment at or near the R1 growth stage (Figure 5), which is in agreement with findings by other investigators [21,22]. The strong relationship of NDRE near the R1 stage with yield (Figure 8) highlights the importance of this remote sensing index for monitoring maize N fertility in the context of predicting profitability. Detailed analyses shown in Figure 6a–d and Figure 7a–d suggested that the low NDRE values observed in maize crops receiving less nitrogen fertilizer and those affected by extreme water content (Figure 6a–d) were due to a reduced chlorophyll content, as inferred by constraining NDRE data extraction to within the crop canopy aerial cross-section (Figure 7a–d). Such findings were contributed to by high-precision GCP data determination during UAV photogrammetry processing. Without RTK-based GCPs, different and, to some degree, misleading results were the case (Figure 6e–h and Figure 7e–h). The relatively less obvious impact of RTK-based GCPs’ inclusion in the case of high-N-rate treatments can be explained by the fact that a rather uniform optimum growth condition was promoted in such environments, making any position drifts during NDRE data extraction have less significant consequences. Careful consideration needs to be taken if this phenomenon is used to justify omitting RTK-based GCPs for the application of UAV photogrammetry in high-N-fertility conditions, because the nature of investigations in precision agriculture is to optimize N fertilizer applications to reduce environmentally risky excessive N applications. Given the examples of misleading information from UAV mapping results when high-precision GCPs are omitted during the processing, this study raises a concern and warning regarding interpretation from similar studies omitting precision GCPs in UAV photogrammetry processing [23,24,25].

Further work is needed to examine the implication of using different RTK-based correction processing approaches, such as OPUS Rapid Static (OPUS-RS) instead of OPUS-S. The OPUS-RS has the advantage of requiring less time for GNSS data collection for RTK-based coordinate determinations between 15 m and 2 h, whereas OPUS-S requires longer than 2 h to achieve accurate solutions. It is worthwhile to investigate the implication of the resulting RTK rover coordinate accuracy and reliability, particularly in the context of the investigation of the nutrient responses of maize crops or similar agronomic studies.

## 5. Conclusions

Research outputs in Baton Rouge, LA, confirm that GNSS augmentation with RTK resulted in a significant improvement of positional data bias from 311 to 4 cm. Such positional accuracy translated into a more reliable result of UAV remote sensing mapping with RTK-based GCP data establishment compared to GCP data determined without GNSS augmentation, as shown by agronomic study results involving N responses in maize crops in Bossier City, LA, USA. This study reveals the strong merit of the inclusion of high-precision positioning in UAV photogrammetry in the context of agronomic field crop research. Such a workload-demanding step of establishing and collecting coordinate data of GCPs is a worthwhile effort as it guarantees planimetrically accurate and consistent remote sensing index maps across data acquisition times, and is therefore ideal in the context of monitoring crop stress, such as exemplified in this study. With precision and consistency, the GIS technique and remote sensing data in this study reveal a causal relationship under N-deficiency conditions between reduced canopy cover and photosynthesis capacity. It was not possible to draw such relationships when UAV photogrammetry processing omitted precise GCP data. One shortcoming for the RTK system used in this study (EMLID RS2) is the requirement of internet connectivity between the RTK rover and the controller unit (android device). Even though the real-time signal transmission between the RTK base unit on the RTK rover unit is facilitated by LoRA technology, an internet connection is required between the rover and the android device to allow the rover to receive kinematic corrections from the base. This problem, however, is only an issue when there is a lack of internet connectivity in the area where the technology is applied. Nonetheless, the EMLID support forum provided a work-around to allow RTK base and rover connection with no internet connection.

## Figures and Tables

**Figure 1 sensors-24-05322-f001:**
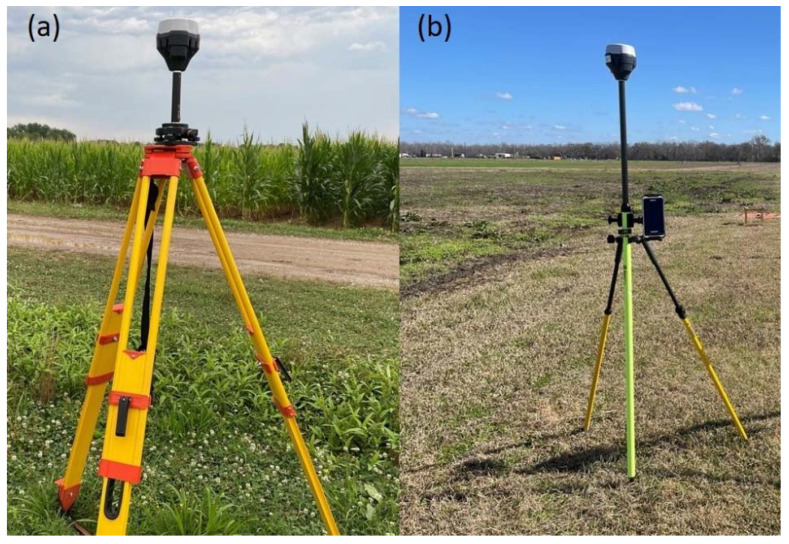
EMLID RS2 RTK system used in this study, showing the base unit mounted on a survey tripod via a tribrach attachment plate (**a**) and the rover unit mounted on the survey rod stabilized with a survey dipod (during stand-by mode), with the Nautiz X6 Android device attached on the survey rod (**b**).

**Figure 2 sensors-24-05322-f002:**
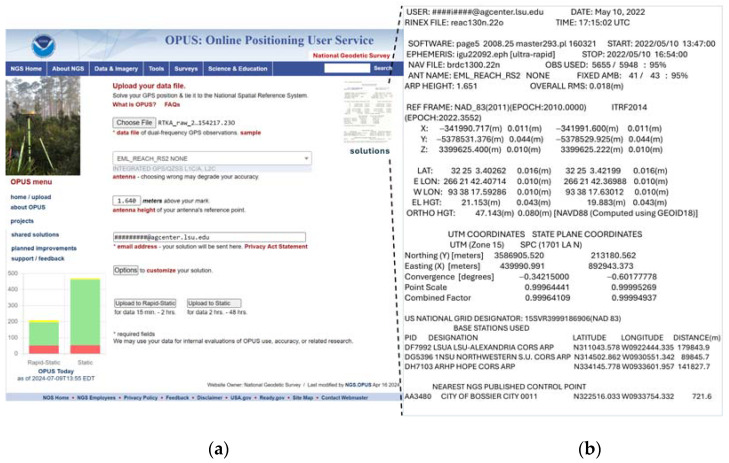
The online submission page of the Online Positioning User Service (OPUS) from the US National Oceanic Atmospheric Administration (NOAA) (**a**) and the resulting coordinate solution retrieved via email for the RTK base station at the LSU AgCenter Red River Research Station in Bossier City, LA, USA (**b**).

**Figure 3 sensors-24-05322-f003:**
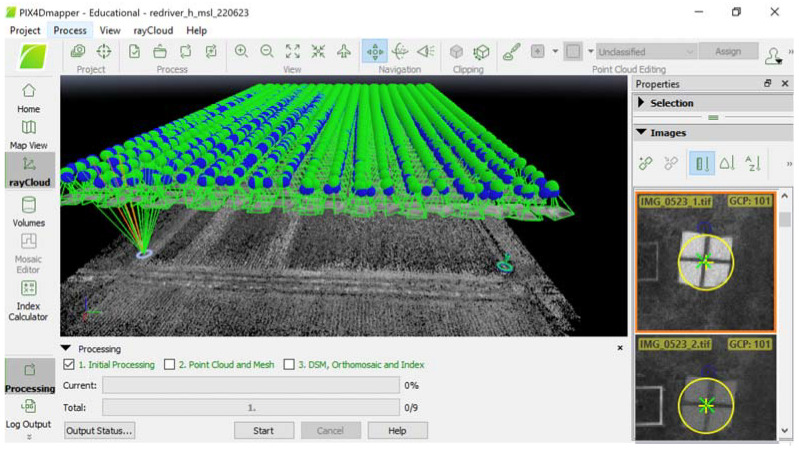
Registered GCP data in Pix4DMapper during initial processing step involving UAV raw camera data. Green and yellow symbol on the photos of GCP markers indicate registered coordinates of the points.

**Figure 4 sensors-24-05322-f004:**
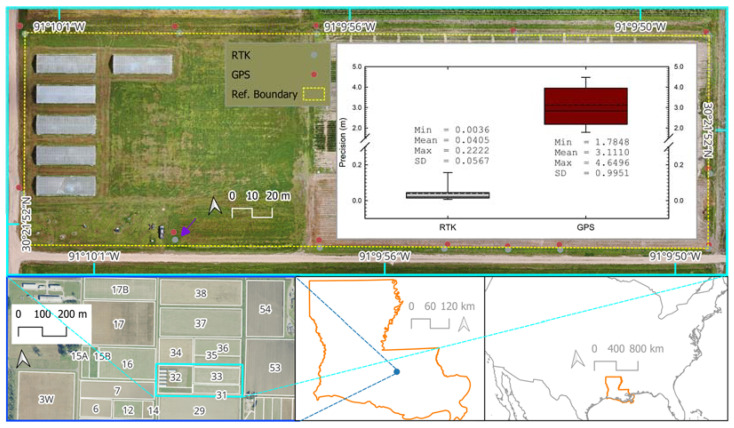
Comparison of planimetric precision of RTK versus GPS coordinate establishment based on evaluation at the Central Research Station, Baton Rouge, LA, USA, on 9 November 2023 (geographic location within Louisiana is shown by the blue symbol in the state level overview map). The purple arrow in the main map points to the 13^th^ assessment point that was not aligned with the boundary line of the block (yellow dashed line). The histogram whiskers represent the 25th and 75th percentiles of the computed distance data between the test and reference points whereas the solid and dashed lines in the histogram represent mean and median precision values, respectively. A background UAV-based RGB map was based on aerial data acquired on 6 June 2023. The source of the background imagery overviewing the Central Research Station was the United States Department of Agriculture (USDA) National Agriculture Imagery Program (NAIP) (image date of 11 November 2021).

**Figure 5 sensors-24-05322-f005:**
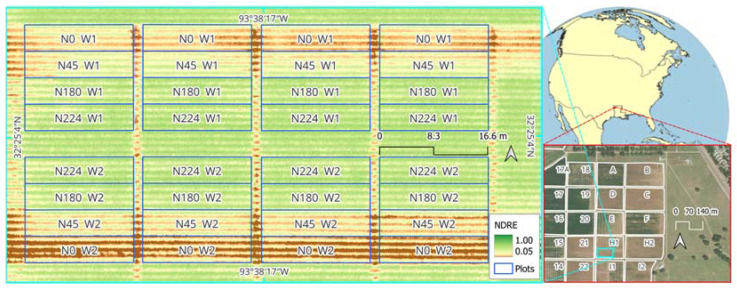
Aerial map of Normalized Difference Red-Edge Index (NDRE) for the maize N rates and excessive water experiment on 3 June 2022 (67 DAP) at block H1 of the LSU AgCenter Red River Research Station, Bossier City, LA, USA. The study location (block H1) is highlighted in cyan in the overview map of the research station, shown with the background aerial imagery map from the United States Department of Agriculture (USDA) National Agriculture Imagery Program (NAIP) (image date of 13 August 2019). The location of the station is indicated by the red symbol in the global overview map.

**Figure 6 sensors-24-05322-f006:**
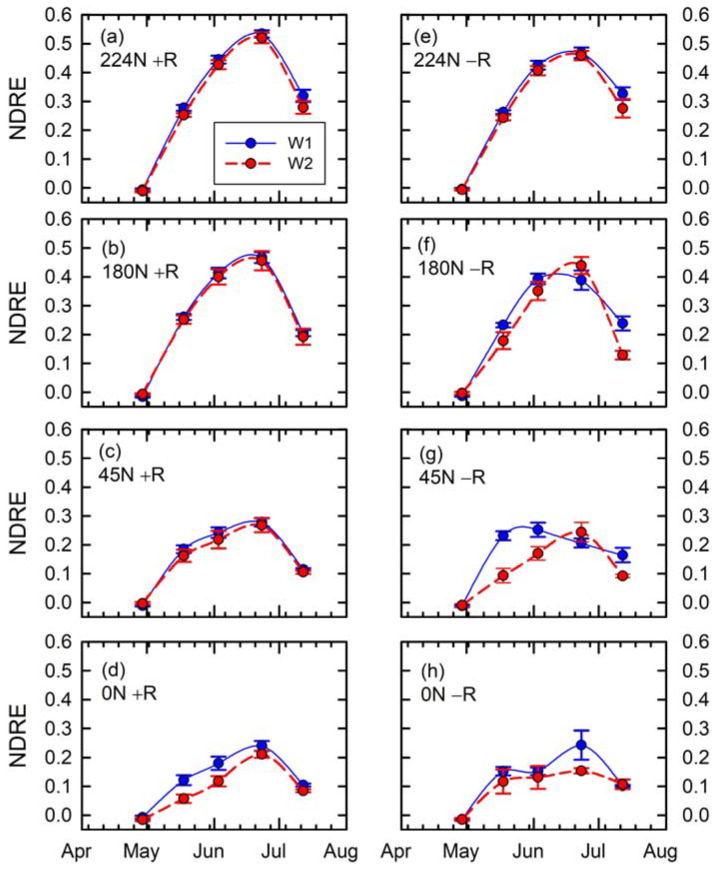
Time series of extracted NDRE data based on entire plot boundaries by N rates (225 kg ha^−1^ in (**a**,**e**), 180 kg ha^−1^ in (**b**,**f**), 45 kg ha^−1^ in (**c**,**g**), and 0 kg ha^−1^ in (**d**,**h**)), water treatments (W1 in blue and solid line for control and W2 in red and dashed line for excessive), and with (**left**, +R) or without (**right**, −R) RTK-based GCPs.

**Figure 7 sensors-24-05322-f007:**
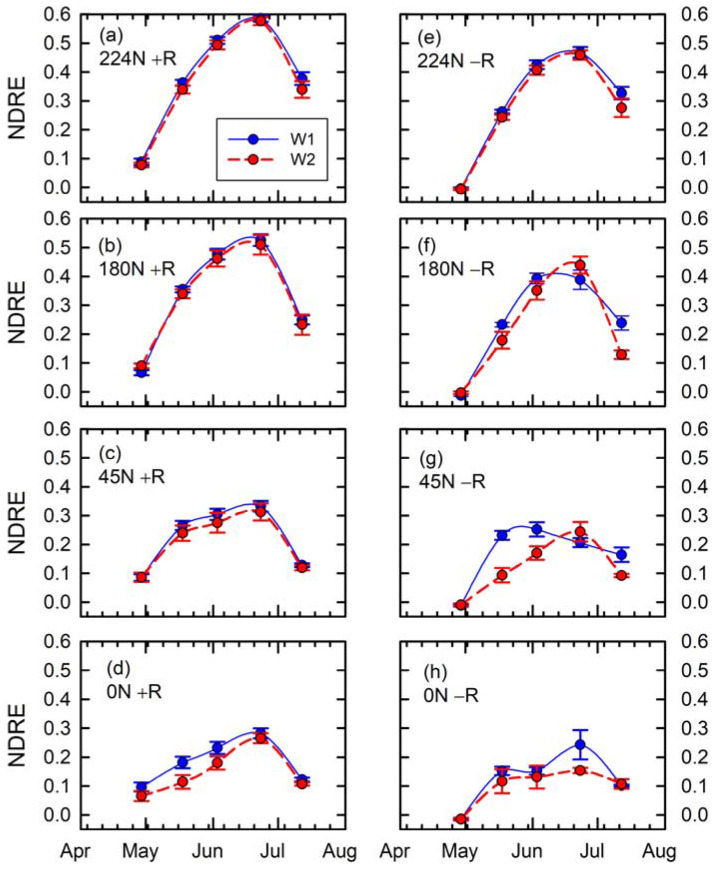
Time series of extracted NDRE data based on narrow polygons along crop rows by N rates (224 kg N ha^−1^ in (**a**,**e**), 180 kg N ha^−1^ in (**b**,**f**), 45 kg N ha^−1^ in (**c**,**g**), and 0 kg N ha^−1^ in (**d**,**h**)), water treatments (W1 in blue and solid line for control and W2 in red and dashed line for excessive), and with (**left**, +R) or without (**right**, −R) RTK-based GCPs.

**Figure 8 sensors-24-05322-f008:**
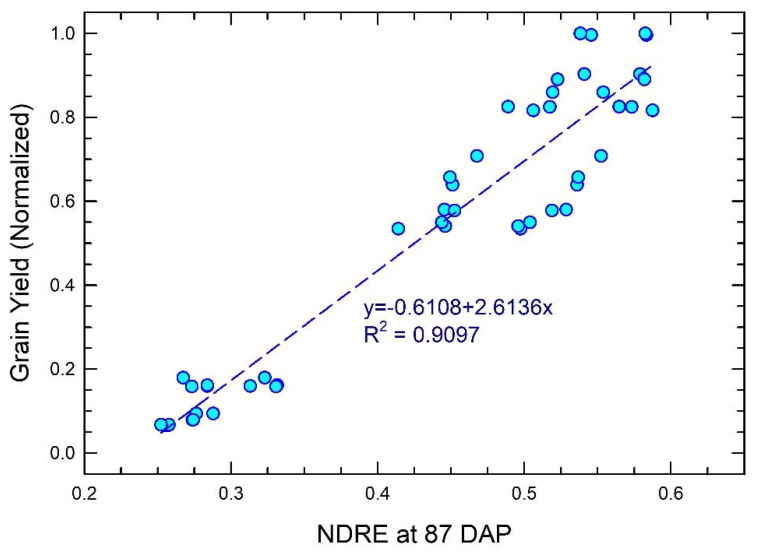
Relationship between NDRE at 87 DAP with maize yield in this study. Data are shown as light blue circles. Blue dashed line represents regression curve labeled with the equation and R^2^ value.

**Table 1 sensors-24-05322-t001:** Comparison of geographic coordinate bias obtained with RTK versus GPS in this study and the literature.

Case		RTK			GPS		Geographic Location
	x (cm)	y (cm)	xy (cm)	x (cm)	y (cm)	xy (cm)	
This study	5.92	3.32	4.05	115.74	304.19	311.11	Bossier parish, LA, USA
Reference study 1 [18]	6.03	5.17	8.70	146.17	54.10	164.10	Gyeongsangnam-do, Republic of Korea
Reference study 2 [19]	*	*	10.00	*	*	*	Marsfield, New South Wales, Australia
Reference study 3 [20]	1.94	2.40	*	*	*	*	Henryk Jordan Park, Krakow, Poland

* Data not reported.

## Data Availability

The original contributions presented in this study are included in the article. Further inquiries can be directed to the corresponding author.

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
