# Peer review of "Precise Positioning in Nitrogen Fertility Sensing in Maize (Zea mays L.)"

_sensors, 2024, doi:10.3390/s24165322_

Round 1

Reviewer 1 Report

Comments and Suggestions for Authors

See attached detailed review

Comments on the Quality of English Language

Pretty good. Just need to re-check for accurate terms

Author Response

Comments 1:

This research is very good and important to understand the role of RTK correction to GNSS measurement to increase the positional accuracy of the object.

Response 1:

Thank you for appreciating and understanding the importance of this study to provide a scientific report on the merit of GNSS-augmentation for agricultural research.

Comments 2:

It is highly recommended to present the detailed methods in post processing the GNSS observation data or even compare different approach in the post processing.

Response 2:

Thank you for the recommendation to present detailed methods in post processing of the GNSS observation data in the context of geodetic GNSS data processing using OPUS NOAA service. Brief descriptions of OPUS processing have been included in the revised manuscript (line # to#). The determination of RTK base position with OPUS in this study was pre-processing (not post-processing) and was a requirement to establish reliable absolute position of the positional data collected by the RTK rover. Comparing different pre-processing options with OPUS, e.g. OPUS-Static versus OPUS-Rapid Static is beyond the scope of this study, however, it is an interesting idea and thus this idea is mentioned in the revised manuscript in the discussion section regarding potential future work (line # to #).

Comments 3:

It is recommended to insert the coordinates of this orthomosaic as well as the vector map or even the inzet maps

Response 3:

Thank you for pointing this out regarding presenting coordinate data in the maps in Figure 4 and 5. This is a very good suggestion. Figure 4 and 5 in the revised manuscript are now with the requested coordinate data presented as grid mark in the main maps with latitude and longitude reference data.

Comments 4:

It is recommended to provide a matrix presenting the results of this research compared with other similar studies to see the difference and improvement

obtained in this study

Response 4:

I am very grateful to the reviewer for this constructive comment. Table 1 has been added in the revised manuscript providing comparison of coordinates data bias obtained in this study against other studies.

Comments 5:

Try to pick some other research with similar settings and pose the similar or even contradictive results as comparison

Response 5:

I am thankful for this comment. Table 1 contain results of coordinates bias against reference data obtained in this study compared to results reported in the literature from different geographic locations.

Reviewer 2 Report

Comments and Suggestions for Authors

This paper investigated precise positioning in N fertility sensing in maize. The topic is interesting. However, it is evident that there have been some severe limitations in the paper:

(1) In abstract section, what method did the authors propose to describe this problem? It is unclear. In addition, the authors pointed out that the RTK system provides cm level precision, could you give the quantitative description?

(2) In introduction section, the problem description is not convincing. The authors should express the purpose of the paper accurately. The authors just listed some related works. The authors should analyze them and introduce the proposed model and methods based on them.

(3) In materials and methods section, section 2.1 and 2.2 are more like literature review. In this section, the authors should address detailed materials and methods like field distribution in maize, dataset collection and processing, and precise positioning methods.

(4) In conclusions section, the authors should give experimental conclusions and shortcomings.

Comments on the Quality of English Language

Minor editing of English language required

Author Response

Comments 1:

In abstract section, what method did the authors propose to describe this problem? It is unclear. In addition, the authors pointed out that the RTK system provides cm level precision, could you give the quantitative description?

Response 1:

Thank you for pointing out the lack of problem description in the abstract. This has been corrected in the revised manuscript. Likewise, the quantitative information of the RTK precision has been added in the revised manuscript.

Comments 2:

In introduction section, the problem description is not convincing. The authors should express the purpose of the paper accurately. The authors just listed some related works. The authors should analyze them and introduce the proposed model and methods based on them.

Response 2:

Thank you for a valuable point regarding making problem statements more convincing in the introduction section. The manuscript now has been revised to make the rationale more compelling why this study is needed, essentially realizing without augmentation GNSS positioning solution contain high degree of uncertainties and it is likely to impact when such in accurate positioning data went on to be used in UAV-based mapping.

Comments 3:

In materials and methods section, section 2.1 and 2.2 are more like literature review. In this section, the authors should address detailed materials and methods like field distribution in maize, dataset collection and processing, and precise positioning methods.

Response 3:

Thank you for pointing this out. Field experiment description for the Baton Rouge, LA site is listed in line # to # for the straightforward RTK vs GPS positioning assessment and for the Bossier City, LA site in line # to # for the detailed assessment involving use of RTK and GPS positioning data in UAV photogrammetry. Additional description of processing of precise positioning involving OPUS-S has been added in the revised manuscript (line # to #).

Comments 4:

In conclusions section, the authors should give experimental conclusions and shortcomings.

Response 4:

Thank you for this valuable comment regarding experimental conclusion and shortcomings. Experimental conclusions have been added and made clearer in the conclusion section. Likewise, one shortcoming has been added in the conclusion section.

Round 2

Reviewer 2 Report

Comments and Suggestions for Authors

agree

Comments on the Quality of English Language

OK